# Insights into the Cellular Localization and Functional Properties of TSPYL5 Protein

**DOI:** 10.3390/ijms25010039

**Published:** 2023-12-19

**Authors:** Sergey A. Silonov, Eugene Y. Smirnov, Eva A. Shmidt, Irina M. Kuznetsova, Konstantin K. Turoverov, Alexander V. Fonin

**Affiliations:** Laboratory of Structural Dynamics, Stability and Folding of Proteins, Russian Academy of Sciences, St. Petersburg 194064, Russia; e.smirnov@incras.ru (E.Y.S.); evashmidt28@gmail.com (E.A.S.); imk@incras.ru (I.M.K.); kkt@incras.ru (K.K.T.)

**Keywords:** TSPYL5, intrinsically disordered proteins (IDPs), intrinsically disordered regions (IDRs), protein–protein interactions, fluorescence recovery after photobleaching (FRAP), liquid–liquid phase separation (LLPS)

## Abstract

In recent years, the role of liquid–liquid phase separation (LLPS) and intrinsically disordered proteins (IDPs) in cellular molecular processes has received increasing attention from researchers. One such intrinsically disordered protein is TSPYL5, considered both as a marker and a potential therapeutic target for various oncological diseases. However, the role of TSPYL5 in intracellular processes remains unknown, and there is no clarity even in its intracellular localization. In this study, we characterized the intracellular localization and exchange dynamics with intracellular contents of TSPYL5 and its parts, utilizing TSPYL5 fusion proteins with EGFP. Our findings reveal that TSPYL5 can be localized in both the cytoplasm and nucleoplasm, including the nucleolus. The nuclear (nucleolar) localization of TSPYL5 is mediated by the nuclear/nucleolar localization sequences (NLS/NoLS) identified in the N-terminal intrinsically disordered region (4–27 aa), while its cytoplasmic localization is regulated by the ordered NAP-like domain (198–382 aa). Furthermore, our results underscore the significant role of the TSPYL5 N-terminal disordered region (1–198 aa) in the exchange dynamics with the nucleoplasm and its potential ability for phase separation. Bioinformatics analysis of the TSPYL5 interactome indicates its potential function as a histone and ribosomal protein chaperone. Taken together, these findings suggest a significant contribution of liquid–liquid phase separation to the processes involving TSPYL5, providing new insights into the role of this protein in the cell’s molecular life.

## 1. Introduction

Intrinsically disordered proteins and proteins with intrinsically disordered regions (IDPs and IDRs) play an important role in the biogenesis of membrane-less organelles formed as a result of biopolymer phase separation, and thus in cellular processes, providing flexible and dynamic regulation of various interactions [1]. Such proteins are also involved in pathological processes and are potential therapeutic targets [2,3].

The intrinsically disordered testis-specific Y-encoded-like protein 5 (TSPYL5) is a member of the TSPY-like protein family, which contains a common predicted nucleosome assembly protein (NAP)-like domain. The TSPYL family was first identified in 1998 [4] and currently consists of six members (from TSPYL1 to TSPYL6) [5]. Due to the presence of the NAP-like domain, it is hypothesized that TSPYL proteins may participate in chromatin remodeling and transcriptional regulation, thereby being involved in numerous cellular processes [6]. It is worth noting that the TSPYL genes, except for TSPYL2, lack introns, which suggests the presence of constitutive biological functions for these genes [5]. In a recent study, it was shown that TSPYL5 preferentially interacts with histone H3/H4 through its NAP-like domain in vitro both in immunoblotting and histone deposition assays [7]. However, the role of this in the cell processes remains unexplored.

The gene encoding the TSPYL5 protein is located on chromosome 8 (8q22.1, NCBI Gene ID: 85453) and is frequently amplified in breast cancer, which has led to considering TSPYL5 as an independent marker of poor prognosis in this type of cancer [8,9]. It has been shown that in breast cancer with a poor prognosis, TSPYL5 promotes the degradation of the tumor suppressor p53 by directly inhibiting ubiquitin-specific peptidase 7 (USP7)—the p53 deubiquitinase [9]. The degradation of p53 associated with the formation of the TSPYL5/USP7 complex has also been demonstrated in lung cancer cells that overexpress one of the potential regulators of TSPYL5–MUC16, which likely induces resistance to cisplatin and gemcitabine in this cell type [10]. It is suggested that decreased expression of the TSPYL5 protein may serve as an indicator of prostate tumor progression and lead to reduced drug sensitivity in prostate carcinoma [5,11]. Additionally, TSPYL5 is associated with the resistance of non-small-cell lung cancer (NSCLC) cells to γ-irradiation through the regulation of the PTEN/AKT signaling pathway [12]. It has been shown that at least phosphorylation of the T120 residue of TSPYL5 is responsible for the cyclic signal transduction of AKT/TSPYL5/PTEN and TSPYL5-mediated regulation of cancer stemness [8]. It is worth noting that peptides based on the PEGylated sequence of TSPYL5_113–125_ (TS120) have shown a strong impact on metastatic abilities (invasion/migration), sphere-forming capacity, and colony formation ability of A549 lung cancer cells, making TSPYL5 a potential target for the treatment of oncological diseases [8].

It is known that TSPYL5 plays an important role in the survival of cancer cells with alternative lengthening of telomeres (ALT). In ALT-positive cells, the telomere elongation process is accompanied by the formation of PML bodies (PML-NBs) associated with the alternative telomere lengthening process (ALT-associated PML-NBs, APBs) [13]. TSPYL5 may be a component of PML bodies and prevent polyubiquitination of POT1 by inhibiting USP7, which acts as an activator of the POT1 E3 ubiquitin ligase [14]. Additionally, TSPYL5 is significantly conjugated with SUMO1 [8], and PML bodies are known to be sites for SUMOylation of many proteins [15]. Accordingly, TSPYL5 is proposed as a therapeutic target for ALT-positive oncological diseases [14].

Despite the aforementioned findings, the role of the TSPYL5 protein in cellular processes remains understudied. Furthermore, its cellular localization is not sufficiently characterized, and conflicting information exists regarding its predominantly cytoplasmic localization [16] versus nuclear localization [14]. To investigate the cellular localization of TSPYL5, this study employed the TSPYL5-EGFP fusion protein, enabling the visualization of both cytoplasmic and nuclear (nucleolar) localization for the first time. A more in-depth examination of TSPYL5-EGFP fragments revealed the role of specific TSPYL5 regions responsible for such localization. Additionally, we investigated the ability of TSPYL5-EGFP and its fragments to undergo phase separation using the fluorescence recovery after photobleaching (FRAP) method. Furthermore, a bioinformatics analysis of the TSPYL5 interactome was conducted, providing insights into potential interactions and functions of this protein in the cell.

## 2. Results and Discussion

### 2.1. TSPYL5 Localization

The TSPYL5 protein can be localized in the cytoplasm and the Golgi apparatus, according to the Human Protein Atlas website (www.proteinatlas.org, accessed on 25 October 2023). These data were obtained using antibodies (HPA031347, Sigma-Aldrich, St. Louis, MO, USA) targeting a region within the ordered NAP-like domain of TSPYL5 in the 268–389 amino acid (aa) region. Immunohistochemistry of TSPYL5 in human kidneys also revealed predominantly cytoplasmic localization, utilizing antibodies targeting the NAP-like domain region 263–302 aa (ab203657, Abcam, Cambridge, UK) [16]. However, such localization did not explain the substantial amount of TSPYL5 detected in the nuclear fraction of cells during co-immunoprecipitation [8]. Using antibodies against the N-terminal region of TSPYL5 (sc-98186, Santa Cruz, CA, USA), nuclear localization was demonstrated [14], specifically co-localization of this protein in PML bodies and in unknown nuclear clusters. However, the study did not detect TSPYL5 localization in the cytoplasm, which was previously shown with the same antibodies [8]. It should be noted that antibodies against TSPYL5 (sc-98186, Santa Cruz) in another study [17] primarily showed co-localization in the cytoplasm, with partial localization observed in the nucleoplasm and potentially the nucleolus.

Since the NAP-like domain of TSPYL5 is represented by the region 198–380 aa and is known to predominantly bind to histones H3/H4 [7], it can be hypothesized that in the formation of a TSPYL5 protein complex, its detection may become challenging for antibodies targeting the NAP-like domain due to steric factors.

To obtain TSPYL5 localization data, we engineered the chimeric protein TSPYL5-EGFP construct for transient transfection. It was demonstrated that TSPYL5-EGFP localized in the cytoplasm and nucleolus of the HeLa cell line (Figure 1A). Moreover, rare events in cells included weak or excessively high intensity of TSPYL5-EGFP in the nucleolus (Figure 1A, panels a1 and a4, respectively). 

An analysis of the acquired images revealed the localization of TSPYL5-EGFP in the nucleolus as a granular component, one of its functions being pre-ribosomal assembly [19] (Figure 1C). 

Given that TSPYL5-EGFP is localized in both the cytoplasm and the nucleolus, we decided to investigate whether this localization is cell-cycle-dependent. For this purpose, a HeLa cell line with a constant expression of the Fucci System for visualizing the cell cycle was generated (Figure 1D). Surprisingly, the localization of TSPYL5-EGFP in the nucleolus remained consistent regardless of the cell cycle phase (Figure 1B). This suggests the existence of a shared cellular TSPYL5 function, independent of the cell cycle.

### 2.2. TSPYL5 Structure and Fragment Analysis

To determine the amino acid sequence regions of TSPYL5 responsible for its cytoplasmic and nucleolar localization, a bioinformatics analysis of the protein sequence was performed. It was found that the TSPYL5 sequence contains two potential nuclear localization signal (NLS) sites: (i) 4–27 aa region with a probability of 0.99; (ii) 132–148 aa region with a probability of 0.79 (Figure 2A). Additionally, there may be two DNA-binding sites (6–22 aa, 121–131 aa) and one RNA-binding site (12–19 aa). It is worth noting that RNA plays a crucial role in the liquid–liquid phase transition, promoting condensation and the formation of liquid droplets inside the cell [20]. 

An analysis of the disorder and propensity of the TSPYL5 protein for spontaneous phase separation using the RIDAO online platform [21], as well as FuzDrop [22] and PSPredictor [23], revealed that TSPYL5 consists of 65.7% intrinsically disordered regions (IDRs), is potentially prone to LLPS, and may act as a driver of phase separation. The main droplet-promoting regions (DPRs) are represented by the regions 1–50, 80–204, and 382–417 amino acids. This result is particularly interesting in the context of a new perspective on the nucleolus as a multi-phase liquid condensate, where LLPS processes play a crucial role [24]. It should be noted that TSPYL5 may have two O-linked glycan sites according to the GlyGen (https://glygen.org/, accessed on 25 October 2023): O-N-acetylglucosamine at T409 and S414. This suggests that the disordered C-terminus of TSPYL5 potentially controls the dynamic cycle of this protein depending on nutrients and stress [25].

In the following step, we performed a cellular localization analysis of TSPYL5-EGFP fragments (Figure 2C). It was shown that the TSPYL5 fragment 1–50 aa exhibited nuclear localization, while the fragment 50–417 aa was not localized in the nucleus, and the 31–50 aa fragment showed a uniform distribution throughout the cell. These results confirm the bioinformatics prediction analysis of the 4–27 aa region of the nuclear localization signal (NLS) site and suggest the absence of an NLS site in the 132–148 aa region. A comparison of the localization of TSPYL5 fragments 1–198 and 1–382 aa clearly demonstrated that the addition of the NAP-like domain (198–382 aa) to the disordered sequence of 1–198 aa led to cytoplasmic localization and reduced protein nucleoplasm localization (Figure 2C). A similar effect was observed when comparing the fragments 50–198 and 50–417 aa, where the addition of the NAP-like domain to the uniformly distributed cytoplasmic 50–198 aa fragment resulted in predominantly cytoplasmic localization of the 50–417 aa fragment of TSPYL5. 

Considering the evident localization of the TSPYL5 fragment 1–50 aa in the nucleolus, we analyzed the TSPYL5 sequence using the NoD predictor of nucleolar localization sequences (NoLSs) in proteins [26]. The conducted analysis revealed a NoLS in the TSPYL5 1–30 aa region (Appendix A), which fully coincides with our data obtained from fluorescence microscopy of TSPYL5-EGFP fragments (Figure 2C).

These observations indicate the following: (i) TSPYL5 has a fine-tuned control element for localization between the cytoplasm and nucleolus; (ii) NLS/NoLSs in the 4–27 aa region are responsible for nuclear (nucleolar) localization of TSPYL5; (iii) the NAP-like domain (198–382 aa) is responsible for the cytoplasmic localization of TSPYL5.

The TSPYL5 sequence was also analyzed for the presence of nuclear export signals (NESs) using LocNES [27] and NLSdb [28] predictors (Appendix A). NLSdb identified one potential NES in the ordered NAP-like domain (282–287 aa). LocNES identified eight potential NESs, with six located in the NAP-like domain. All eight sites exhibited low score values, and the highest score (0.243) was found in the 169–183 aa region. The corresponding fragment of TSPYL5 50–198 aa demonstrated uniform distribution throughout the cell (Figure 2C), indicating the absence of NESs in this region. The presence of NESs in the ordered TSPYL5 NAP-like domain remains unknown.

### 2.3. FRAP Analysis of TSPYL5 and Its Fragments

The exchange dynamics of TSPYL5-EGFP and its fragments between nucleoli and nucleoplasm were investigated. EGFP in the nucleoli was completely bleached, and the rate and extent of fluorescence recovery after photobleaching (FRAP) were evaluated. The obtained fluorescence recovery curves of TSPYL5-EGFP showed a nearly complete overlap between the full-length TSPYL5 and its disordered fragment 1–198 aa (Figure 3A). The immobile fraction accounted for approximately 50%, and the half-time of recovery was approximately 3.5 s, leading to the following findings: (i) the disordered region 1–198 aa significantly contributes to the dynamics of TSPYL5 exchange with the nucleoplasm; (ii) the immobile fraction of TSPYL5 is associated with the disordered region 1–198 aa, suggesting that this region may be involved in immobilizing interactions with protein (or RNA) partners; (iii) the 3.5 s half-time of recovery indicates a potential role of liquid–liquid phase transition in this process [29]. It is worth noting that the disordered fragment of TSPYL5 1–50 aa has an immobile fraction of approximately 15% and a half-time of recovery of approximately 1 s, further demonstrating its partial role in the aforementioned findings.

### 2.4. TSPYL5 Interactome Analysis

We analyzed the interactome of the TSPYL5 protein, consisting of 65.7% of intrinsically disordered regions (IDRs) and prone to spontaneous phase transition, as a potential driver of liquid–liquid phase separation (LLPS) (Section 2.2). Within the TSPYL5 interactome (from BIOGRID v4.4 database, accessed on 2 November 2023), we identified 49 human proteins (Figure 4A) and 1 SARS-CoV-2 protein—Nucleoprotein. The identification of the SARS-CoV-2 Nucleoprotein was accomplished through proximity labeling mass spectrometry, as reported in two studies investigating the SARS-CoV-2 interactome [31,32]. 

It is known that Nucleoprotein packages the viral genome RNA into a helical ribonucleocapsid (RNP) and plays a fundamental role in virion assembly through interactions with the viral genome and membrane protein M [33]. It is worth noting that nuclear localization is a common feature of coronavirus nucleoproteins, and this protein may disrupt host cell division [34]. Recently, it has been shown that SARS-CoV-2 induces the expression and cytoplasmic translocation of the nucleolar protein nucleolin (NCL), which interacts with SARS-CoV-2 viral proteins and co-localizes with the N protein in the nucleolus and stress granules [35]. This observation may indicate a yet-unexplored role of TSPYL5 in the lifecycle of SARS-CoV-2.

The disorder and propensity for LLPS analysis of 49 TSPYL5 protein partners revealed a high proportion of intrinsically disordered proteins (89.8%) and proteins with disordered regions (10.2%), as well as a high percentage of both driver LLPS (61.2%) and client LLPS (30.6%) (Figure 4B). This result implies a potential significant contribution of LLPS to the molecular processes involving TSPYL5 and its partners.

The GO analysis of the 49 TSPYL5 proteins identified (Figure 4A) that TSPYL5 potentially participates in cytoplasmic translation (GO:0002181), chromatin remodeling (GO:0006338), and regulation of DNA-templated transcription (GO:0006355). The molecular function primarily encompasses RNA binding (GO:0003723). Among the analyzed proteins that interact with TSPYL5 according to BIOGRID, three major groups were distinguished based on their functional properties, making a substantial contribution to the results of the GO analysis (Figure 4A): histones (six proteins), RNA helicases (three proteins), and ribosomal proteins (six proteins). Below is a brief summary of the literature information on these groups.

#### 2.4.1. TSPYL5 and Histones

In vitro experiments have shown that TSPYL5 acts as a histone chaperone responsible for histone H3/H4 deposition and nucleosome assembly [7]. However, it is worth noting that the results obtained from the GST pull-down assay in the same study show weak binding with αH2A and αH2B during immunoblotting.

It is known that half of the H3 and H4 histones incorporated into chromatin in each daughter cell are recycled from old tetramers and heterodimers, while the rest are newly synthesized structures [36]. Newly synthesized H3 and H4 histones are also incorporated into chromatin throughout the cell’s lifespan during various DNA-related processes, such as transcription [37]. After synthesis, histones bind to histone chaperones, a class of proteins that bind and transport histones, preventing their chaotic aggregation during nucleosome formation [38,39]. Two pathways for histone import into the nucleus are proposed: the first involves the transfer of H3-H4 histone dimers (Imp4, ASF1), and the second, recently supported by experimental evidence, suggests a model of import into the nucleus as monomers (with the help of the protein Imp5) [37].

It remains unclear how the levels of nuclear histones are regulated, whether histones undergo degradation, and by what mechanisms these processes occur. In a recent study [40], it was found that histone H2B primarily undergoes degradation through the proteasomal pathway, with lysine-120 of H2B playing a crucial role in its K48-linked polyubiquitination and subsequent degradation. Inhibition of the degradation process resulted in an increased distribution of H2B in nucleoli [40]. This suggests that nucleoli participate in the regulation of H2B histone degradation. 

The role of TSPYL5 as a histone chaperone and its cellular localization in the cytoplasm and nucleolus (Figure 1A) are of particular interest. On one hand, they may suggest its potential function as a histone transporter into the nucleus, and on the other hand, they may indicate its potential role as a histone chaperone simultaneously in the cytoplasm and nucleolus. It is noteworthy that TSPYL5 fragments with the NAP-like domain but lacking the disordered N-terminal region (50–417, 198–417, 198–382 aa) exhibit cytoplasmic localization. Therefore, TSPYL5, potentially bound to histones in the cytoplasm through its NAP-like domain, exhibits nuclear translocation only when its disordered N-terminal region (1–50 aa) is present. A more detailed investigation of this process in the future will provide insights into the underlying mechanisms.

#### 2.4.2. TSPYL5, Ribosomal Proteins, and RNA-Helicases

It is known that specialized chaperones protect newly synthesized ribosomal proteins (r-proteins) from aggregation and escort them during assembly into forming ribosomes. Recently, it has been discovered that the protein Nap1 (NAP-family member), in addition to its main function, can act as such a chaperone [41]. The precedent of interaction between a NAP-family representative and ribosomal proteins supports the hypothesis that TSPYL-5 can function not only as a histone chaperone but also as a chaperone for ribosomal proteins due to the presence of a NAP-like domain in its structure. It should be noted that the NAP-like domain of TSPYL5 lacks the extended additional region found in most NAP family proteins in various species [7], and it is replaced by a disordered region of unknown function.

Within the TSPYL5 interactome, three representatives of DEAD-box RNA helicases were identified using the affinity capture mass spectrometry (AC-MS) method: DDX6 [42], DDX21, and DDX24 [43] (Figure 4A). 

It is known that DEAD-box RNA helicases are important regulators of RNA metabolism and are involved in cancer development [44]. These helicases constitute the major repetitive family of RNA-binding proteins that are crucial for genome protection [45]. Recently, it has been shown that DDX21 occupies the transcribing rDNA locus in the nucleolus; directly interacts with both rRNA and miRNA; and facilitates the transcription, processing, and modification of rRNA [46]. It is known that among the DEAD-box RNA helicases, DDX6 is involved in a wide range of RNA life processes: translation inhibition, mRNA degradation, transport and storage, translation activation, and nuclear export [45]. For DDX24, a correlation has been shown between its elevated levels in hepatocellular carcinoma tissues and poor prognosis of this disease [47]. Interestingly, DDX24 is necessary for the organization of muscle fibers and specification of the anterior pole, which is important for regenerating the head in planarians [48].

## 3. Materials and Methods

### 3.1. Plasmids

The fluorescent TSPYL5-EGFP construct used for transient transfection was based on the pTag-GFP2-C vector (Evrogen, Moscow, Russia). The TSPYL5 gene was isolated using a cDNA obtained by reverse transcription of mRNA isolated from the blood of a healthy donor. The isolated genes were amplified using corresponding primers and then cloned into the pTag-GFP2-C vector through restriction–ligation. TSPYL5-EGFP fragments were obtained by PCR using Q5 polymerase (NEB, Ipswich, MA, USA) with the appropriate primers. The pUltraHot-Fucci construct was amplified from the Fucci reporter pCCL-CellCycle plasmid and cloned into the pUltraHot vector using restriction–ligation, in which the mCherry gene was replaced with the puromycin resistance gene. The pCCL-CellCycle plasmid was a gift from John M. Greally (Addgene plasmid #132429), and the pUltraHot plasmid was a gift from Malcolm Moore (Addgene plasmid #24130). Plasmids were prepared using electroporation and Escherichia coli DH5 alpha cells. Several clones were selected for sequencing. Plasmid isolation was performed using the Plasmid Miniprep kit (Evrogen, Russia). All constructs were sequenced using the BigDye Terminator v3.1 Cycle Sequencing Kit (Thermo Fisher Scientific, Waltham, MA, USA), and the resulting samples were analyzed using the ABI PRISM 3500 genetic analyzer (Applied Biosystems, Foster City, CA, USA).

### 3.2. Cell Culture and Stable Cell Line

HeLa and HEK293T cell lines were kindly provided by the cell culture collection of the Institute of Cytology RAS. The cell lines were cultured in DMEM medium (Biolot, Saint-Petersburg, Russia) supplemented with 10% FBS (Cytiva, Marlborough, MA, USA), L-glutamine (Biolot, Saint-Petersburg, Russia), and penicillin–streptomycin (Biolot, Saint-Petersburg, Russia). The cells were maintained at 37 °C and 5% CO_2_ in a humidified incubator.

The Fucci cell line construction was performed using a lentiviral vector system. HEK293T cells at 70% confluency were co-transfected with the pUltraHot-Fucci, psPAX2, and pMD2.G plasmids using PEI MAX (Polysciences, Warrington, PA, USA) as a transfection reagent. The psPAX2 plasmid was a gift from Didier Trono (Addgene plasmid #12260), and the pMD2.G plasmid was a gift from Didier Trono (Addgene plasmid #12259). After 48 h of incubation at 37 °C, 5% CO_2_, the supernatant was collected, centrifuged at 300× *g* for 5 min, filtered through a 0.45 µm PES filter unit, and added to the HeLa cell line. After 24 h, the medium was changed, and a selective antibiotic was added for 5 days. After selection, the Fucci-stable cell line was confirmed using a fluorescent microscope.

### 3.3. Confocal Microscopy and Fluorescence Recovery after Photobleaching (FRAP)

Live cell imaging was performed by plating cells on 35 mm glass Ibidi dishes. After 24 h, cells were transfected with plasmids using GeneJect 39 (Molecta, Moscow, Russia). The cells were visualized by irradiating them with a laser at wavelengths of 488 nm, 565 nm, or 641 nm to excite EGFP, RFP, or BFP fluorescence, respectively. The resulting fluorescence was recorded using an Olympus FV3000 confocal microscope (60 × oil immersion objective, NA 1.42). Images were corrected for background signal and high-frequency noise using ImageJ software (Version 1.53c).

The analysis of protein dynamics was performed using the fluorescence recovery after photobleaching (FRAP) technique. Initially, five images of an object were obtained, and then the object was photobleached by irradiating it with a 488 nm laser for 5 s at a power of 10 mW. Subsequently, fluorescence recovery images of the object were acquired for 2 min. Image analysis was performed using ImageJ software (Version 1.53c). Curves were generated and analyzed using Prism 7 (GraphPad Software, San Diego, CA, USA).

### 3.4. Prediction and GO Analysis

The prediction of the nuclear localization site (NLS) was performed using a hidden Markov model based on a two-state model, utilizing NLStradamus [49]. The prediction of DNA- and RNA-binding regions was conducted using ProNA [50]. Intrinsic disorder analysis, liquid–liquid phase separation (LLPS) prediction, and the determination of biological processes and molecular functions of proteins were performed as outlined in [51]. Briefly, information on the TSPYL5 protein interactome was sourced from the BIOGRID database (https://thebiogrid.org/, accessed on 21 September 2023). A total of 49 proteins were selected from the database for analysis, utilizing various online tools and predictors. Intrinsic disorder prediction was carried out using the RIDAO online platform [22], which integrates multiple predictors employing different algorithms. Furthermore, the probability of proteins undergoing liquid–liquid phase separation was analyzed using the FuzDrop [23] and PSPredictor [24] predictors. A protein was assigned to the “LLPS” category if it had a PSPredictor score > 0.5 and a FuzDrop score > 0.6. In cases where there were conflicting results between predictors, the protein was labeled as “Controversial LLPS”. The definition of biological processes and molecular functions of proteins was performed using the Enrichr resource (https://maayanlab.cloud/Enrichr/, accessed on 21 September 2023), and the 10 terms with the lowest *p*-values were selected. See Appendix A.

## 4. Conclusions

The use of full-length TSPYL5-EGFP and its parts allowed the visualization of TSPYL5 both in the cytoplasm and the nucleus simultaneously for the first time. It was also revealed that in nucleus, the majority of the protein is located in the nucleolus.

The investigation of TSPYL5-EGFP fragments has demonstrated that the disordered N-terminal region plays a significant role in nuclear/nucleolar localization and the protein’s exchange dynamics. On the other hand, an ordered NAP-like domain assumes a crucial role in TSPYL5 cytoplasmic localization.

Literary and bioinformatics analyses suggest a potential role of TSPYL5 as a chaperone for histones and ribosomal proteins. Of particular interest is the interplay between these protein functions and the dual localization of TSPYL5, raising the possibility of TSPYL5 acting as a chaperone–transporter for histones and ribosomal proteins. Importantly, the obtained data on the exchange dynamics of TSPYL5 with the nucleoplasm, along with the bioinformatics analysis of the disorder and propensity for liquid–liquid phase separation (LLPS) of the TSPYL5 protein and its partners, have led to the hypothesis of a significant contribution of liquid–liquid phase separation to processes involving TSPYL5.

In summary, our results present opportunities for future research on TSPYL5 and its partners, providing new insights into the role of this protein in the molecular network of cellular activities.

## Figures and Tables

**Figure 1 ijms-25-00039-f001:**
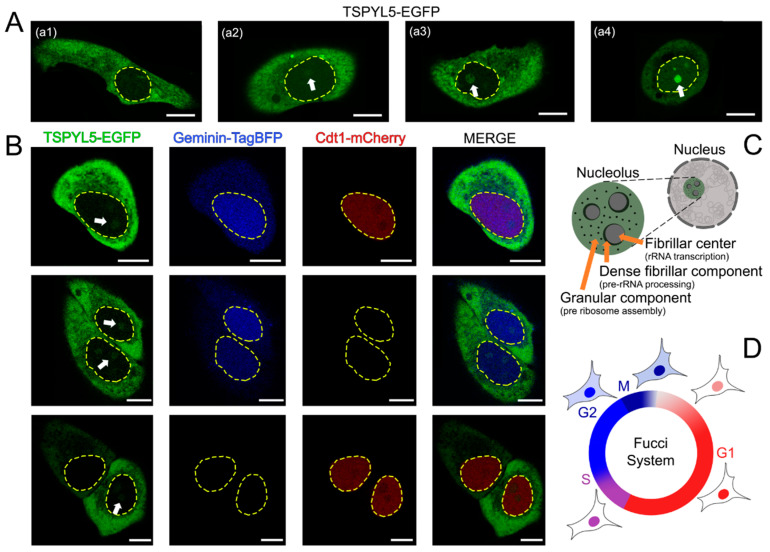
TSPYL5-EGFP localization. (**A**) Localization of the chimeric protein TSPYL5-EGFP in the cytoplasm and nucleolus of HeLa cells. The fluorescent signal intensity in the nucleolus may vary among different cells in the same cell line (**a1**–**a4**). (**B**) Presence of nucleolar localization at different stages of the HeLa cell cycle using the Fucci system. (**C**) Simplified structure of the nucleolus, showing the fibrillar center, dense fibrillar component, and granular component. Green color highlights the data from fluorescent microscopy showing the localization of TSPYL5-EGFP in the granular component of the nucleolus. (**D**) Model illustrating the working principles of the Fucci system for cell cycle analysis [18]: Cdt1-mCherry expression is observed during the G1 phase, while Geminin-TagBFP expression is suppressed; both proteins are present during the S phase; Geminin-TagBFP is expressed and Cdt1-mCherry expression is suppressed during the G2 and M phases. Yellow dashed line indicates the cell nucleus; white arrow indicates the nucleolus. Scale bar 10 μm. The illustrations in (**C**,**D**) were created with BioRender.com (accessed on 10 September 2023).

**Figure 2 ijms-25-00039-f002:**
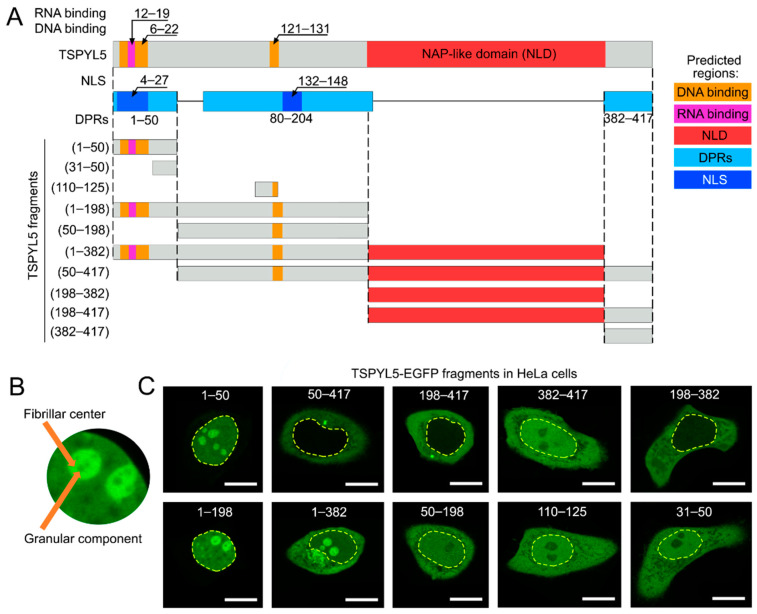
TSPYL5 structure and fragment analysis. (**A**) Schematic representation of the TSPYL5 amino acid sequence and its fragments with the results of bioinformatics analysis (details in the Materials and Methods section). DNA-binding regions (orange), RNA-binding regions (pink), NAP-like domain (red), droplet-promoting regions (DPRs, cyan), and nuclear localization signal (NLS, blue) are shown. (**B**) Enlarged image of the TSPYL5-EGFP fragment 1–198 aa localization in the putative nucleolus granular component. (**C**) TSPYL5-EGFP fragment localization in HeLa cell line. The presence of nuclear and nucleolar localization is shown for TSPYL5 fragments 1–50 aa, 1–198 aa, and 1–382 aa. The numbers represent the positions of amino acid residues (aa). The yellow dashed line indicates the cell nucleus. Scale bar 10 µm.

**Figure 3 ijms-25-00039-f003:**
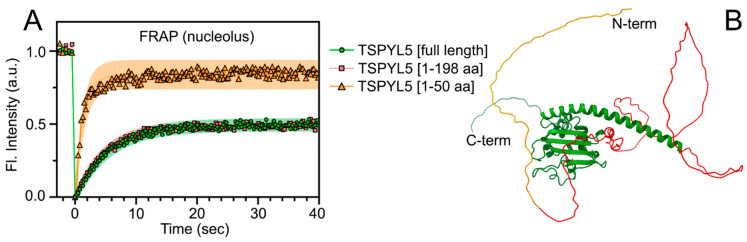
Fluorescence recovery after photobleaching (FRAP) of full-length TSPYL5-EGFP and its fragments. (**A**) FRAP recovery curves. The red curve representing the recovery of the 1–198 aa fragment closely overlaps with the green curve representing full-length TSPYL5. (**B**) Three-dimensional structure of TSPYL5 according to AlphaFold2 prediction (AF-Q86VY4-F1) [30]. Regions of TSPYL5 are highlighted: 1–50 aa (yellow), 50–198 aa (red), and 198–417 aa (green).

**Figure 4 ijms-25-00039-f004:**
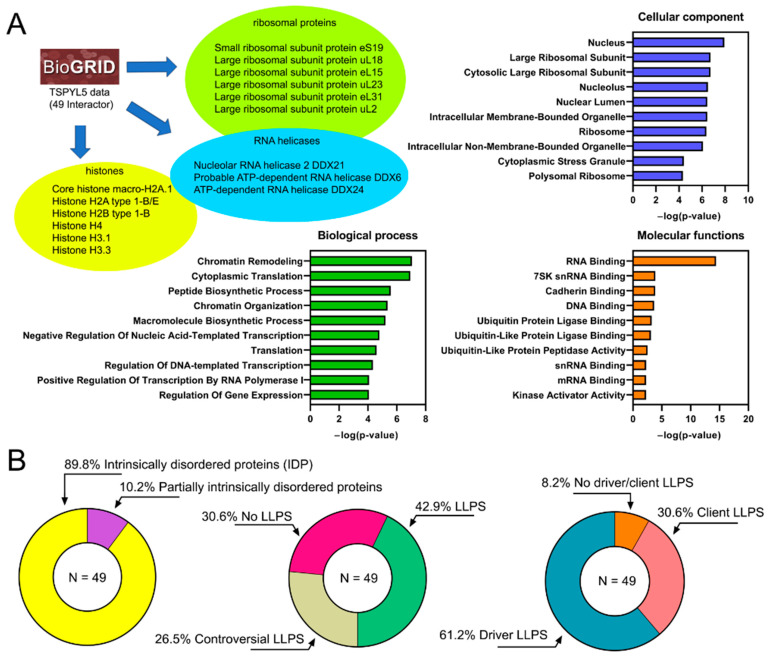
Analysis of human TSPYL5 interactome. (**A**) Three major blocks of the TSPYL5 interactome based on their functions and frequent occurrence in the database (histones, RNA helicases, ribosomal proteins). GO analysis for the 49 proteins in the interactome is presented, including cellular components, molecular functions, and biological processes. (**B**) Results of the analysis of TSPYL5 interactome for disorder, potential propensity for LLPS, and the role of the protein in LLPS, shown as pie charts.

## Data Availability

Data are contained within the article and Appendix A.

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
