# Peer review of "Insights into the Cellular Localization and Functional Properties of TSPYL5 Protein"

_ijms, 2023, doi:10.3390/ijms25010039_

Round 1

Reviewer 1 Report

Comments and Suggestions for Authors

In the paper “Insights into the cellular localization and functional properties of TSPYL5 protein” the authors report their experimental data on two different topics: the experimental cellular localization of the TSPYL5 protein and a bioinformatics analysis of the protein sequence and of the putative interactions with its partner.

The results of the localization investigation are consistent. The structure and fragment analysis was performed using some bioinformatics platforms and the results confirmed by FRAP. The following TSYPL5 interactome exploration, conducted again by a bioinformatic approach, was also quite interesting.

I am not really enthusiast of papers that are made of a consistent part of predictions, even if the questioned algorithms are several and different. Nevertheless, the data are consistent and the paper is well written. It can be published on International Journal of Molecular Science.

Comments on the Quality of English Language

Minor editing of English language required

Author Response

We are very grateful to Reviewer 1 for positive evaluation and high rating of our work. Minor editing of English language was done

Reviewer 2 Report

Comments and Suggestions for Authors

Testis-specific Y-encoded protein (TSPY)–like protein 5 (TSPYL5) is a protein containing a nucleosome assembly protein (NAD)-like domain, described as a histone chaperone.

The first part of the article is dedicated to the analysis of the subcellular localization of TSPYL5. The authors performed an analysis of TSPYL5 protein structure in order to determine the regions responsible for nuclear and nucleolar localization. They constructed the chimeric TSPYL5-EGFP and TSPYL fragments-EGFP to investigate their localization. Since the full-length protein is mainly localized in the cytoplasm. I wondered why the authors did not look for the presence of potential NES sequences. For comforting their results, the authors should mutate functional NoLSs and NLSs (and NES) in the full-length protein and analyze the localization of the TSPYL5-EGFP mutants.

               The second part is an analysis of the interactome of TSPYL5 form the BIOGRID database. The authors identified 1 SARS-CoV-2 protein (line 199, Figure 4A). I couldn’t find the SARS-CoV-2 protein – Nucleoprotein in Figure 4A as indicated line 199. Could the authors put the context of the studies which revealed this interaction, site the articles and indicate the technical used for determining the interaction. Same comments for DDX21, DDX6 and DDX24.

Conclusion: the authors should moderate their claims. I do not think that the nucleolus localization of TSPYL5 in predominant since Figure 1A showed that TSPYL5-EGFP is mainly localized in the cytoplasm in 3 out of the 4 images shown.

Author Response

We are very grateful to Reviewer 2 for positive evaluation of our work and interesting ideas.

-The first part of the article is dedicated to the analysis of the subcellular localization of TSPYL5. The authors performed an analysis of TSPYL5 protein structure in order to determine the regions responsible for nuclear and nucleolar localization. They constructed the chimeric TSPYL5-EGFP and TSPYL fragments-EGFP to investigate their localization. Since the full-length protein is mainly localized in the cytoplasm. I wondered why the authors did not look for the presence of potential NES sequences. For comforting their results, the authors should mutate functional NoLSs and NLSs (and NES) in the full-length protein and analyze the localization of the TSPYL5-EGFP mutants.

Reply: We utilized a varied suite of NLS/NES predictors; however, NES was not detected, and only NLS was reported in this study. Following Reviewer 2 comments, we supplemented the article with results from LocNES and NLSdb predictors, which demonstrated some NES outcomes among the NLS/NES predictions.

The LocNES predictor identified 8 potential NES regions, yet six of them reside in the NAP-like domain, and all eight sites have low prediction scores (< 0.25). Among the eight potential NES according to LocNES, the highest prediction score (0.243) was founded with the 169-183 aa region. However, our cellular experiments demonstrated that the TSPYL5 50-198 aa fragment exhibits uniform distribution throughout the cell, indicating the absence of NES in this region. NLSdb identified a single "potential" site "VEELGL" (282-287 aa) corresponding to the NAP-like domain of TSPYL5.

An analysis of NES prediction has been added to the article.

We believe that making point mutations in the TSPYL5 NAP-like domain without a detailed understanding of the interaction between TSPYL5 and its partners through that only-one ordered domain would be methodologically unsound. Such mutations may impact the correct folding of the protein's singular ordered domain, potentially leading to unreliable data on the protein's localization.

In response to Reviewer 2 comment regarding the Nuclear Localization Signal (NLS), our approach in this study was grounded in the notion that, in order to conduct experiments on NLS sites, it is imperative to first identify these regions harboring NLS. As delineated, our analysis identified two potential NLS regions: 4-27 aa (score 0.99) and 132-148 aa (score 0.79). While we had hoped to experimentally verify both NLS sites, only the fragment TSPYL5 (1-50 aa) exhibited nuclear and nucleolar localization. Accordingly, we have experimentally confirmed that the NLS/NoLS is located within the TSPYL5 1-50 aa region. It is noteworthy that predictors based on the amino acid sequence identify the region of 4-27 aa with a high prediction score of 0.99 (equivalent to 99%). Thus, in our study, we conclude that TSPYL5 4-27 aa serve as the NLS/NoLS. It is important to emphasize that the 4-27 aa region corresponds to both NLS and NoLS predictors (as observed experimentally with expression in both the nucleus and nucleolus). To elucidate the precise mechanism of NLS or NoLS, a comprehensive Western blot analysis with potential NLS and NoLS partners, followed by the point mutations analysis validating mechanism, is necessary. However, these highly time and resource-intensive tasks were beyond the scope of the current study.

               - The second part is an analysis of the interactome of TSPYL5 form the BIOGRID database. The authors identified 1 SARS-CoV-2 protein (line 199, Figure 4A). I couldn’t find the SARS-CoV-2 protein – Nucleoprotein in Figure 4A as indicated line 199. Could the authors put the context of the studies which revealed this interaction, site the articles and indicate the technical used for determining the interaction. Same comments for DDX21, DDX6 and DDX24.

Reply: We have made corrections to the text and to Figure 4A.

The TSPYL5 interactome, according to the BIOGRID database, consists of 49 human proteins and 1 SARS-CoV-2 protein (resulting in a total of 50 proteins). Figure 4A depicts only the 49 human proteins as 'interactor TSPYL5.' Since considering only 1 SARS-CoV-2 protein did not provide meaningful insights, it was not included in Figure 4A. However, this protein has also been analyzed using a bioinformatics approach, and the data are available in the supplementary material, Table S1.

In response to Reviewer 2 comment, we have included references to four specific proteins of interest in the text of our paper, directly retrieved from the BIOGRID v4.4 database.

-Conclusion: the authors should moderate their claims. I do not think that the nucleolus localization of TSPYL5 in predominant since Figure 1A showed that TSPYL5-EGFP is mainly localized in the cytoplasm in 3 out of the 4 images shown.

Reply: In accordance with the Reviewer 2 comment, the Conclusion section has been revised as follows: «The use of full-length TSPYL5-EGFP and its parts allowed to visualize TSPYL5 both in cytoplasm and nucleus simultaneously for the first time. It was also revealed that in nucleus the majority of protein is located in nucleolus».

Round 2

Reviewer 2 Report

Comments and Suggestions for Authors

The authors answered to my concerns. It's a shame that the authors did not consider the mutation of NoLSs and NLSs useful. I agree that such mutations may impact the correct folding of the protein, as do the addition of GFP-tag. However, these experiments could have really reinforced the message.